# `UnfoldML`: Cost-Aware and Uncertainty-Based Dynamic 2D Prediction for Multi-Stage Classification

**Yanbo Xu**[1,*]**, Alind Khare**[1,*]**, Glenn Matlin**[1]**, Monish Ramadoss**[1]**,
Rishikesan Kamaleswaran**[2]**, Chao Zhang**[1]**, Alexey Tumanov**[1]
[1]Georgia Institute of Technology,
[2] Emory University
Atlanta, GA

## Abstract

Machine Learning (ML) research has focused on maximizing the accuracy of predictive tasks. ML models, however, are increasingly more complex, resource intensive, and costlier to deploy in resource-constrained environments. These issues are exacerbated for multi-stage prediction where stages transition in a progression with "happens-before" relationship.We argue that it is possible to "unfold" a monolithic single multi-class classifier, typically trained for all stages using all data, into a series of single-stage classifiers. Each single-stage classifier can be then cascaded gradually from cheaper to more expensive classifiers, which are trained using only the necessary data modalities or features required for that stage. Hence, we propose `UnfoldML`, a cost-aware and uncertainty-based dynamic 2D prediction pipeline for multi-stage classification that enables (1) navigation of the accuracy/cost tradeoff space, (2) reduction in spatio-temporal cost of inference, and (3) early prediction on proceeding stages. `UnfoldML` achieves orders of magnitude better cost in clinical settings, while detecting multi-stage disease development in real time. It achieves within 0.1% accuracy from the highest-performing multi-class baseline, while saving close to 20X on spatio-temporal cost of inference and earlier (3.5hrs) disease onset prediction. We also show that `UnfoldML` generalizes to image classification, where it can predict different level of labels (from coarse to fine) given different level of abstractions of a image, saving close to 5X cost with as little as 0.4% accuracy reduction.

## 1 Introduction

Machine Learning (ML) research has mostly focused on improving prediction accuracy for classification tasks, such as image classification (Foret et al., 2020; Xie et al., 2017), disease risk prediction (Feng et al., 2020; Xu et al., 2018), pedestrian detection (Zhang et al., 2018; Cai et al., 2015), etc. The understandable drive for high accuracy has often resulted in deeper, complex neural networks, which can incur high memory (*spatial* cost) and high latency (*temporal* cost) at inference time. However, the deployment of ML applications must be cost-aware. Run time environment like mobile devices or bedside-patient monitors are commonly resource constrained, and applications that can be offloaded to cloud computing always aim for a reduced cloud bill. In this paper, we focus on developing a pipeline that can balance between high prediction accuracy and low *spatio-temporal* cost in deploying ML classification models.

We consider a scenario of deploying ML classifiers in a multi-stage classification task where one predicted class can progressively transform to a next stage of classes, characterized as "happens-before" relationship between classes. This task is commonly observed in many real-world applications. For instance in clinical settings, disease progression is often identified by a series of stage transitions

---

*Authors contributed equally to this research.

36th Conference on Neural Information Processing Systems (NeurIPS 2022).

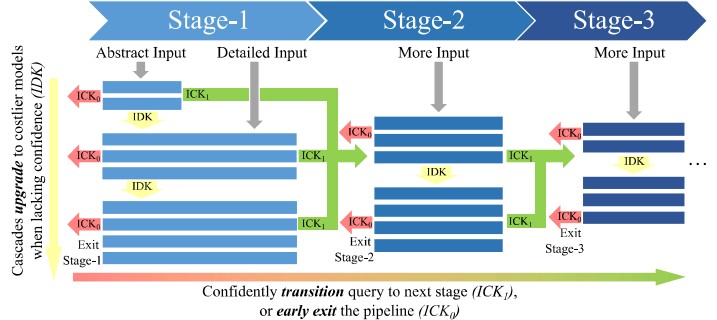 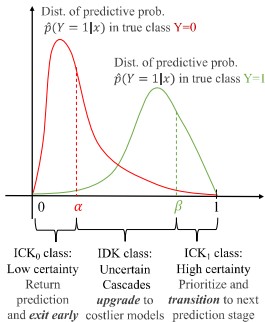

(a) The 2D query propagation mechanism in `UnfoldML`

(b) $IDK$ and $ICK$ classes

Figure 1: 2D uncertainty-based propagation in `UnfoldML`: Queries that are in confidently low risk will return $ICK_0$ and be monitored by cheaper models; queries that are hard to predict will return $IDK$ and be advanced to costlier but more confident models; queries that are in confidently high risk will return $ICK_1$ and be transited to next stage.

(Sperling et al., 2011; Singer et al., 2016). Detection on early stage of the disease allows doctors to take appropriate actions in time before it enters into late severe stages. In image classifications, recognition from super classes to sub classes in a coarse-to-fine manner has shown improved classification performance (Dutt et al., 2017; Lei et al., 2017). In all these applications, general multi-class classifiers (Liu et al., 2019; Fagerström et al., 2019; Foret et al., 2020) have been developed by treating all the stages as multi classes and achieved state-of-the-art prediction performance, but at significantly high spatio-temporal cost. Prior work (Wang et al., 2017) trades off accuracy for low cost but still ignores the key relationship between the classes, so it fails to find the optimal trade-off.

We propose `UnfoldML`[2] : a cost-aware and uncertainty-based prediction pipeline for dynamic multi-stage classification. It "unfolds" a monolithic multi-class classifier into a series of single-stage classifiers, reducing its deployment cost. Each single-stage classifier is then cascaded gradually from cheaper to more expensive binary classifiers, further reducing the cost by dynamically selecting an appropriate classifier for an input query. Figure 1 summarizes the two dimensional (2D) query propagation *mechanism* designed in `UnfoldML`: Horizontally it allows a query to transition through multiple stages, and vertically it allows the query to progressively upgrade to costlier models constrained by the pre-specified budget limit. It computes a classifier's prediction confidence on a query then directs the query through one of the following three gates: 1) "*I confidently know NO*" ($ICK_0$), which rejects the current query and early exits from the pipeline (exit); 2) "*I don't know*" ($IDK$), which upgrades the query to a higher accuracy but costlier model, producing a more confident result within the budget (vertical cascading); and 3) "*I confidently know YES*" ($ICK_1$), which transitions the query to the next stage of the prediction task (horizontal forwarding). The design of $ICK_0$ gate allows queries to early exit from the `UnfoldML` so it reduces the overall spatio-temporal cost of the pipeline. The $ICK_1$ gate allows queries to faster transition to next stages so it enables early prediction on late stages, which can be critical in clinical settings (Reyna et al., 2019). Overall, the combined 2D propagation mechanism uniquely enables the navigation of the cost/accuracy tradeoff space for searching an optimal set of *policies* for dynamic model selection at inference time.

We also propose two training algorithms for learning the optimal *policies* for the designed 2D query propagation in `UnfoldML`. The key idea of the training algorithms is to learn the optimal thresholds on the gating functions $ICK_0$, $IDK$ and $ICK_1$. The first proposed *hard-gating* algorithm assumes the gating functions to be step functions parameterized by deterministic thresholds on prediction confidence. To find the optimal thresholds, it performs bottom-up grid search over a topologically-sorted model list and stops at the cutoffs that minimize the prediction loss subject to the cost constraint. The secondly proposed *soft-gating* algorithm defines the gating functions to be probabilistic activation functions. It follows a Mixture-of-Expert (MoE) framework to adaptively determine a threshold for one model conditionally on all the other model's prediction confidence for the given query. To obviate the cost of running all models in MoE, we further propose a Dirichlet Knowledge Distillation (DKD) to run only a cheap multi-label classifier that is trained for distilling the Bayesian predictive uncertainties of all models.

We summarize the contributions of this paper as follows:

---

[2]Code is available at `https://github.com/gatech-sysml/unfoldml`.

- We design a novel 2D query propagation pipeline that "unfolds" multi-stage prediction workflows by leveraging the "happens-before" relationship between the stages, and achieves a lower-cost prediction pipeline with minimal accuracy degradation.

- We propose two learning algorithms for sufficiently navigating the cost/accuracy tradeoff space and searching for the optimal policies realizing the designed 2D query propagation.

- We apply the proposed pipeline to two real-world applications and demonstrate it reduces the spatio-temporal cost of inference by orders of magnitude.

## 2   Related Work

The most relevant work to our proposed method is the one-step $IDK$ cascade (Wang et al., 2017), which incorporates prior work of "*I don't know*" ($IDK$) classes (Trappenberg and Back, 2000; Khani et al., 2016) into cascade construction and introduce a latency-aware objective into the construction comparing with previous cascaded prediction frameworks (Rowley et al., 1998; Viola and Jones, 2004; Angelova et al., 2015). Another group of work focus on the problem of feature selection assuming each feature can be acquired for a cost. They train a cascade of classifiers for optimizing the trade-off between the expected classification error and the feature cost. Early solution (Raykar et al., 2010) limits the cascade to a family of linear discriminating functions. Cai et al. (2015) applies boosting method for cascading a set of weak learners. Recent methods (Trapeznikov and Saligrama, 2013; Clertant et al., 2019; Janisch et al., 2019) develop POMDP-based frameworks and incorporate deep Q-learning in training the cascades. In contrast to all of the above work that are only 1-D pipelines for one-step prediction task (can be multi-class classifications), our method extends to a 2D pipeline that can dynamically forward examples to next steps after they are confidently predicted as passed on the current step. Further, we also develop a more efficient pipeline framework based on Mixture-of-Experts (MoE) modeling and knowledge distillation, which can apply gradient decent algorithms for learning the parameters efficiently.

The idea of MoE was originally introduced by Jacobs et al. (1991), for partitioning the training data and feeding them into separate neural networks during the learning process. This gate decision design is applied into many domains such as language modeling (Ma et al., 2018), video captioning (Wang et al., 2019), multi-tasking learning (Ma et al., 2018). It is also used in network architecture searching (Eigen et al., 2013) by setting gate activation on network layers. Sparse gates are introduced in MoE so that it can efficiently select from thousands of sub-networks (Shazeer et al., 2017) as well as increases the representation power of large convolutional networks by only using a shallow embedding network to produce the mixture weights (Wang et al., 2020). We incorporate the idea of sparsely gated MoE (Shazeer et al., 2017; Wang et al., 2020) into our prediction framework, and design a soft-gating training algorithm by using ReLU as the sparse gating function and imposing L1-norm regularization on the gating weights for further sparsity.

Confidence criterion has been incorporated into active learning by Li and Sethi (2006) and then extended by Zhu et al. (2010). Lei (2014) proposed confidence based classifiers that identifies the confident region (like $ICK$ class) and uncertain region (like $IDK$ class) in predictions. Confidence are also introduced into word embedding (Vilnis and McCallum, 2015; Athiwaratkun and Wilson, 2018) and graph representations (Orbach and Crammer, 2012; Vashishth et al., 2019). Our method posits thresholds on prediction confidence for activating the gates in pipeline expansion. Bayesian Prior Networks (BPNs) (Malinin and Gales, 2018) have been proposed to estimate the uncertainty distribution in model predictions, which is more computationally efficient than traditional Bayesian approaches (MacKay, 1992; Mackay, 1992; Hinton and Van Camp, 1993). We propose Dirichlet Knowledge Distillation (DKD) based on BPNs for distilling prediction uncertainty in large models so that we only need to run a low-cost multi-head model for producing the weights in MoE efficiently.

## 3   Multi-Stage Dynamic Prediction Pipeline

We introduce a dynamic 2D prediction pipeline `UnfoldML`, which learns optimal policies for making "*I confidently know*" ($ICK$) predictions on sequential multi-stage classification tasks. An optimal policy will effectively trade off prediction accuracy against spatio-temporal costs in order to maximize the overall system accuracy's AUC while staying under user-imposed cost constraints.

## 3.1 Problem Formulation

Given $x \in \mathcal{X}$ at time $t$, a multi-stage pipeline decides whether the individual should maintain at the current stage $s$ or progress into the next stage $s + 1$ for time $t + 1$. If there are a total number of $S$ stages that need to be detected, we train $K_s$ number of models for each specific stage $s$ to form a model zoo $\mathcal{M} = \{\{m_{s1}, ..., m_{sK_s}\}\}_{s=1}^{S}$. We measure each model's spatio-temporal cost by multiplying the device cost per unit time with the serving time per prediction stage, denoted as $cost(m_{sk})$ for any model $m_{sk} \in \mathcal{M}$.

To optimize the limited system resources, we design a 2D `UnfoldML` in the following way: (1) start with the simplest model to predict the initial stage on an incoming data, and (2) **upgrade** vertically to a costlier model if the current model returns "*I don't know*" ($IDK$), or (3) **transition** horizontally to the next stage in the pipeline if the current model returns "*I confidently know YES*" ($ICK_1$), or otherwise (4) **exit** the pipeline if the current model returns "*I confidently know NO*" ($ICK_0$). Figure 1 (a) demonstrates the proposed 2D architecture of `UnfoldML`. The central problem of `UnfoldML` is to learn an optimal policy that specifies the three classes of gating functions with an objective of maximizing the system-wide accuracy while minimizing the overall prediction cost.

We formulate `UnfoldML` as a decision rule mapping function $m^{casc} : \mathcal{X} \times \mathcal{M} \to \mathcal{M}$, which takes the query data $x_t$ coming at time $t$ and the current model choice $m_{sk}$ as inputs and determine whether the model for the query should take one of the aforementioned three actions: *upgrade* vertically, *transition* horizontally, or *exit* the pipeline. These decisions rules can be realized by two groups of parameters: a confidence criterion $q : \mathcal{X} \times \mathcal{M} \to [0, 1]$ that measures the confidence score of a model's prediction on a query data, and two gate functions $G^{IDK}, G^{ICK_1} : [0, 1] \times \{\text{True}, \text{False}\}$ that are applied to the defined confidence score. The two exclusive gates $IDK$ and $ICK_1$ each respectively decides if the current prediction belongs to an $IDK$ class such that (s.t.) the system will *upgrade* the query to a costlier model but still remaining within the user-defined cost budget, or an $ICK_1$ class s.t. the system will *transition* the query to the next stage in the pipeline. The third gate $G^{ICK_0}$ is then determined as $\neg G^{IDK} \wedge \neg G^{ICK_1}$. Formally, we can write the decision rule used at each prediction stage as follows

$$m^{casc}(x_t, m_{sk}; q, G) = \begin{cases} m_{s(k+1)}, & G^{IDK} \wedge \neg G^{ICK_1}(q_{sk}(x_t)), \\ m_{(s+1)1}, & \neg G^{IDK} \wedge G^{ICK_1}(q_{sk}(x_t)), \\ m_{sk}, & \neg G^{IDK} \wedge \neg G^{ICK_1}(q_{sk}(x_t)) \end{cases} \tag{1}$$

where $q_{sk}(x_t)$ is a short notation for $q(x_t, m_{sk})$ measuring the confidence of model $m_{sk}$'s prediction on data $x_t$. The goal of configuring an optimal pipeline given a restricted computation resource can be formalized as the following optimization problem:

$$\min_{G} \mathcal{L}(m^{casc}; \mathcal{D}) \quad \text{s.t. } cost(m^{casc}; \mathcal{D}) \le c, \tag{2}$$

where $G$ consists of the two gate functions $G^{IDK}$ and $G^{ICK_1}$, $\mathcal{L}$ denotes the end-to-end prediction loss on data $D$ and $c$ is a user-specified cost-constraint for the system.

## 3.2 Gate Parameters Learning

Given a training data set (it should be a different set from the data that was used for training the model zoo $\mathcal{M}$) $\mathcal{D} = \{(x_i, (y_i^1, t_i^1), \cdots, (y_i^S, t_i^S))\}_{i=1}^{N}$, where $x_i = (x_{i1}, \cdots, x_{iT_i})$ is an input sequence observed for individual $i$, $y_i^s \in \{0, 1\}$ indicates whether the individual entered to stage-$s$, and if *yes*, we use $t_i^s \in \emptyset \cup [1, T_i]$ to denote the time index when it entered. We first partition the multi-stage data into $S$ one-stage data sets: $\mathcal{D}^s = \{(x_{i[t_i^{s-1}:t_i^s]}, y_i^s == 1)\} \cup \{(x_{i[t_i^{s-1}:T_i]}, y_i^s == 0)\}$, then divide the learning of $IDK$ and $ICK$ gate parameters into two separable sub-problems:

**Sub-Objective 1:** $\min_{G_s^{IDK}} \mathcal{L}^s(m_s^{casc}; \mathcal{D}^s)$ s.t. $cost(m_s^{casc}; \mathcal{D}^s) \le c_s, s = 1, \cdots, S$ $\qquad$ (3)

**Sub-Objective 2:** $\min_{G^{ICK_1}} \mathcal{L}(m^{casc}; \mathcal{D}, G^{IDK^*})$,

where $\mathcal{L}^s$ is the one-stage prediction loss on data $\mathcal{D}^s$, and $c_s$ is the cost budget that is pre-allocated for stage-$s$ satisfying $\sum_s c_s = c$. Decomposing the end-to-end optimization problem in Eq. (2) into two sub-problems in Eq. (2) allows us to parallelize the training process. We can efficiently learn each stage's optimal $IDK$ gate parameters by solving Sub-Objective 1, and learn the optimal $ICK_1$ gate parameters by fixing the $IDK$ gate parameters as the optimal values $G^{IDK^*}$ learnt from last step and then solving Sub-Objective 2.

### 3.2.1 Hard-gating Training Algorithm

In this algorithm, we assume $G^{IDK}$ to be a *hard-gating* function that is parameterized by a cutoff $\alpha_{sk}$ s.t. the gate is only activated if the level of confidence in the prediction is below the threshold.

$$\text{Hard-gating: } G^{IDK}(q_{sk}(\boldsymbol{x}_t)) = \mathbb{I}(q_{sk}(\boldsymbol{x}_t) < \alpha_{sk}), \tag{4}$$

where $\mathbb{I}(\cdot)$ is an indicator function. Based on the problem definition in Eq. (1), a model $m_{sk}$ at stage $s$ can only be activated if $\mathbb{I}(q_{sk}(\boldsymbol{x}_t) \geq \alpha_{sk}) \wedge_{j=1}^{k-1} \mathbb{I}(q_{sj}(\boldsymbol{x}_t) < \alpha_{sj}) \equiv 1$. So we can write the conditional probability of being at stage-$s$ given an input query $\boldsymbol{x}_t$ as

$$\Pr(y^s = 1 | \boldsymbol{x}_t; m_s^{\text{casc}}) = \sum_{k=1}^{K_s} \mathbb{I}(q_{sk}(\boldsymbol{x}_t) \geq \alpha_{sk}) \cdot \prod_{j=1}^{k-1} \mathbb{I}(q_{sj}(\boldsymbol{x}_t) < \alpha_{sj}) \cdot m_{sk}(\boldsymbol{x}_t),$$

where $m_{sk}(\boldsymbol{x}_t) = \Pr(y^s = 1 \mid \boldsymbol{x_t}; m_{sk})$. The Sub-Objective 1 loss in *hard-Gating* algorithm is defined as the negative log-likelihood loss

$$\mathcal{L}_{\text{nll}}^s(m_s^{\text{casc}}; \mathcal{D}^s) = -\sum_{i=1}^{N_s} \sum_{t=1}^{T_i} y_{it}^s \cdot \log p_{it}^s + (1 - y_{it}^s) \cdot \log(1 - p_{it}^s), \tag{5}$$

where $p_{it}^s = \Pr(y^s = 1 | \boldsymbol{x}_{it}; m_s^{\text{casc}})$, $y_{it}^s = 1$ only if $y_i^s = 1$ and $t \in [t_i^1 - \delta_t, t_i^1]$ for some $\delta_t$ time-steps we wish to early detect the next stage $s + 1$.

Algorithm 1 in Appendix 5.1 describes a bottom-up grid search algorithm for learning *hard-gating* parameters. We first sort the model list $\mathcal{M}^s = \{m_{s1}, \cdots m_{sK_s}\}$ in a monotonically increasing order w.r.t both the AUC score and spatial-temporal cost that were provided in model's profiles; this will remove any sub-optimal models. For any stage $s$, Algorithm 1 starts by assigning all the $N^s$ samples for that stage to the first model $m_{s1}$, and performing a grid search on the gate parameters $\alpha_{sk}$'s for level $k = 1$ to $K_s$. It gradually assigns $IDK$ queries to the next level's model in the list until the cost exceeds the user-defined budget $c_s$ for that stage. In each iteration of searching the cutoff $\alpha_{sk}$, we set an upper bound $maxA$ on the maximum searching value to avoid over-upgrading. Without setting this bound, the algorithm could overfit and put all the queries into the $IDK$ class, then assign them all to the next level's model. This consumes the cost quota quickly, and prevents those highly $IDK$ queries from exploring costlier models in the list.

### 3.2.2 Soft-gating Training Algorithm

In contrast to grid search on the thresholds, we propose a *soft-gating* algorithm formulating an objective function that can be efficiently solved using gradient descent algorithms. In this algorithm, we define the gate function $G^{IDK}$ to be a ReLU function parameterized by a pair of coefficients $(a_{sk}, b_{sk})$ s.t. the gate is only activated if the linear product $a_{sk} \cdot q_{sk}(\boldsymbol{x}_t) - b_{sk} > 0$. Formally we define Soft-gating as

$$\text{Soft-gating: } G^{IDK}(q_{sk}(\boldsymbol{x}_t)) = \text{ReLU}(a_{sk} \cdot q_{sk}(\boldsymbol{x}_t) - b_{sk}). \tag{6}$$

Therefore the conditional probability of being at the stage-$s$ given input $\boldsymbol{x}_t$ can be defined as a mixture of the $K_s$ models available for stage-$s$ prediction:

$$\Pr(y^s = 1 | x_t; m^{\text{casc}}) = \sum_{k=1}^{K_s} G^{IDK}(q_{sk}(\boldsymbol{x}_t)) \cdot m_{sk}(\boldsymbol{x}_t) / \sum_{j=1}^{K_s} G^{IDK}(q_{sj}(\boldsymbol{x}_t)). \tag{7}$$

Now, the negative log-likelihood loss $\mathcal{L}_{\text{nll}}^s$ becomes solvable using gradient descent algorithms. However, the normalization term in the mixture of experts requires running all the candidates models in the zoo, which conflicts with our cost-saving goal. Therefore, we propose a Dirichlet Knowledge Distillation (DKD) procedure for training a small surrogate model per each stage to approximate the prediction confidence of a model without truly running it on a query. The smaller distilled model only needs to be run once per query, requiring much less cost than running all the models. In our experiment, we utilize the first model $m_{s1}$ from each stage, take the embedding of $h_{s1}(\boldsymbol{x}_t)$ prior to the last activation layers in $m_{s1}$ and feed it into a 4-layer $K_s$-head Multi-Layer Perceptron (MLP) that is defined as our distilled model in the DKD procedure.

In details, the idea of DKD is to posit a Dirichlet prior distribution over the parameters $\boldsymbol{\pi}$ characterizing the predicted output categorical distribution (i.e., binomial in our setup) and a surrogate prior network $\boldsymbol{f}$ is fit to generate the concentration parameters $\boldsymbol{\alpha}_{sk}$ in the prior:

$$\Pr(\boldsymbol{\pi} | \boldsymbol{x}_t; m_{sk}) = \text{Dir}(\boldsymbol{\pi} | \boldsymbol{\alpha}_{sk}); \quad \boldsymbol{\alpha}_{sk} = (\alpha_{sk,0}, \alpha_{sk,1}) = \boldsymbol{f}(\boldsymbol{x}_t; m_{sk}).$$

If the learnt concentration parameters yield a flat prior distribution, it means high uncertainty in the model prediction; if they yield a sharp prior distribution, it means low uncertainty. Then an estimation $\hat{q}_{sk}(\boldsymbol{x}_t)$ of the confidence score can be computed from the expected predictive probability $\hat{p}_{sk}(\boldsymbol{x}_t) = \mathbb{E}_{\boldsymbol{\pi} \sim \text{Dir}(\boldsymbol{\pi}|\boldsymbol{\alpha}_{sk})}[\pi_1] = \alpha_{sk,1}/(\alpha_{sk,0} + \alpha_{sk,1})$. For training the DKD model, which is defined as a $K_s$-head MLP per each stage in this paper, we write the loss function for head $k$ as the Kullback-Leibler (KL) divergence between the prior distribution and empirical observed distribution:

$$\mathcal{L}(\boldsymbol{\alpha}_{sk}) = \sum_{i=1}^{N_s} \sum_{t=1}^{T_i} \text{KL}\big(\text{Dir}(\boldsymbol{\pi}|\boldsymbol{\alpha}_{sk}) \,\|\, p_{sk}(\boldsymbol{x}_{it})\big),$$

where $p_{sk}(\boldsymbol{x}_{it})$ are the true predicted probabilities produced from model $m_{sk}$ on input $\boldsymbol{x}_{it}$. Additionally, we also add the cross-entropy loss as an auxiliary loss when training the DKD.

Once the distilled model is trained, we can replace the $q_{sj}(\boldsymbol{x}_t)$'s with the estimated values $\hat{q}_{sj}(\boldsymbol{x}_t)$'s in Eq. (7), and rewrite the loss $\mathcal{L}_{\text{nll}}^s$ in Eq. (5) accordingly. Further, we can ensure the selected models do not exceed user-imposed cost constraint by enforcing sparse gating weights over the model choices. Therefore, we define the Sub-Objective 1 loss in *soft-gating* algorithm as

$$\min_{G^{IDK}} \mathcal{L}_{\text{nll}}^s(m^{\text{casc}}; \mathcal{D}_s) + \lambda \mathcal{L}_{\text{cost}}^s + \mu \mathcal{L}_{\text{sparse}}^s,$$

where the second term $\mathcal{L}_{\text{cost}}^s$ takes the cost constraint, controlled by $\lambda > 0$ and the third term $\mathcal{L}_{\text{sparse}}^s$ imposes further sparse regularization on the gating weights w.r.t their $\text{L}_1$ norms, controlled by $\mu > 0$. More specifically, we write the last two terms as $\mathcal{L}_{\text{cost}}^s = \big(\max(0, cost(m_s^{\text{casc}}) - c_s)\big)^2$ and $\mathcal{L}_{\text{sparse}}^s = \sum_{i,t} \|G^{IDK}(\hat{q}_{sk}(\boldsymbol{x}_{it}))\|_1$. Now wecan use stochastic gradient descent algorithms to learn the optimal $\boldsymbol{a}^s$ and $\boldsymbol{b}^s$ minimizing the above loss. Algorithm 2 in Appendix 5.1 summarizes the *soft-gating* training algorithm.

### 3.2.3 Overall Training Algorithm

To complete the overall training algorithm, we need to learn the optimal gate parameter $G^{ICK_1}$ in Sub-Objective 2. Given the model $m_{sk}$ picked by one-stage $IDK$ cascade (either using *hard-gating* or *soft-gating* algorithms to solve Sub-Objective 1) for a data query $\boldsymbol{x}_t$ at stage $s$, we grid search for the optimal thresholds $\theta_{sk}$'s on the predictive probabilities s.t. the type I error on class $ICK_1$ and type II error on $ICK_0$ are both minimized. Several existing methods (Liu, 2012; Perkins and Schisterman, 2006; Unal, 2017; Miller and Siegmund, 1982) have been proposed for minimizing both type I error and type II error in various ways, we pick the *Closet-to-(0,1)* (Perkins and Schisterman, 2006) method that finds the optimal threshold achieving the most left upper corner in the ROC curve. Finally, Algorithm 3 in Appendix 5.1 gives the end-to-end training Algorithm for `UnfoldML`, where we can use either Algorithm 1 or 2 to learn the optimal $IDK$ gating policy.

## 4 Experiments

We evaluate `UnfoldML` on two real-world tasks. The first task is to predict if and when a patient who was newly admitted into the Intensive Care Unit (ICU) of a hospital will develop sepsis (Stage-1), which may worsen and progress into septic shock (Stage-2). The second task is to detect the *subcategory-of-interest*, which utilizes the label hierarchy to first filter out queries that are not in the coarse class of interest in Stage-1 and then refine the predictions into fine classes in Stage-2.

### 4.1 Task 1: Sepsis-Septic Shock prediction

We use MIMIC-III Critical Care Database (Johnson et al., 2016). The database consists of deidentified health records from over $50,000$ critically ill patients who stayed in the ICUs of the Beth Israel Deaconess Medical Center between 2001 and 2012. We detail our data preparation in Appendix 5.2 The final cohort includes a total of $34,475$ ICU patients, from which $2,370$ $(6.8\%)$ presented with Sepsis, from which a total of $229$ $(9.7\%)$ progressed into Septic Shock. We randomly split our cohort of patient data into a training set $(70\%)$, validation set $(20\%)$ and test set $(10\%)$. First, we use the training set to train a set of models to formulate a model zoo. Next, we use the validation set to train the `UnfoldML` policy. Finally, we use the test set to evaluate performance.

### 4.1.1 Experimental Setup

**Model Zoo.** We construct our model zoo by training two sets of binary classifiers for Stage-1 (Sepsis) and Stage-2 (Septic Shock) from a variety of model architectures and input features. We select CPU-based models such as Logistic Regression, Decision Tree and Random Forest, and GPU-based models such as LSTM (Long Short-Term Memory) that is thus far the state-of-art approach for early detection of Sepsis and Septic Shock (Fagerström et al., 2019; Liu et al., 2019). Specifically for LSTM, we vary the hidden size ranging from 100 to 400, the number of layers from 1 to 4. We also vary the input window of patient data from 1, 6, to 12 hours. Furthermore, at each stage we train different model structures on different combinations of input data modalities: all models start with with basic demographic features and vital signs, and then progressively add more features such as laboratory test results, other beside monitoring biomarkers, and medication/IV treatments.

**Confidence Measures** We consider four choices for measuring the confidence $q$ of a model prediction. Given $p$, the model's predictive probability of output $Y$ being 1, we define

- Max probability: $\max\left(1-p, p\right)$,
- Entropy: $\left(p \cdot \log p + (1-p) \cdot \log(1-p)\right)$,
- Entropy of expected: $-\frac{\alpha_0}{\alpha_0+\alpha_1} \cdot \left(\psi(\alpha_0) - \psi(\alpha_1)\right) + \psi(\alpha_0 + \alpha_1)$,
- Mutual Information: Entropy $-$ Entropy of expected,

**Spatio-Temporal Cost.** Given the trained model zoo, we profile spatio-temporal costs for each of the models. The spatio-temporal cost is the total time spent in each hardware due to inference/forward-pass calls of the models (temporal cost) multiplied with the hardware's cost per unit time (spatial cost). This cost serves as a proxy for the real dollar cost as it is the basis for pricing models in cloud offerings such as AWS On-Demand, Lambda, or Spot.

**Baseline.** To our knowledge, `UnfoldML` is the first system to provide 2-dimensional cascading predictions. A reasonable baseline for our method is the set of models that frame the multi-stage sequential task as a multi-class classification task by ignoring the *happen-before* relationship between the stages. The goal of these baseline models becomes classifying patients into one of the three classes: Non-Septic, Septic, and Septic Shock. We take LSTM as our baseline since it is the state-of-art classifier for the task. We evaluate the performance of LSTM's across different spatio-temporal costs by changing the number of layers and hidden sizes in its architecture.

● *Multi-class LSTM*: One unified multi-class model works end to end to predict the multi classes.

Since most of the existing work implements a 1-D prediction on single-stage tasks, we reduce `UnfoldML` to a 1-D pipeline by unfolding it only vertically with no *ICK* transitions and compare the proposed *soft-gating* algorithm with prior works. We evaluate them on the two single-stage classification tasks of predicting sepsis and septic shock respectively.

● *IDK-cascade* (Wang et al., 2017): Prior work of 1-D prediction cascade on *IDK* classes using a hard-gating like algorithm.

● *Single-Stage binary classifiers*: CPU-based models such as Logistic Regression, Decision Tree, Random Forest and TREWScore (Henry et al., 2015), a cox proportional hazards model that is well-known in early detection of septic shock; GPU-based models such as LSTMs.

### 4.1.2 Evaluation and Results

**End-to-End Classification Performance.** To evaluate the prediction performance on our multi-stage task, we treat `UnfoldML` as a multi-class classifier which predicts one of the same three classes as defined in Baseline. `UnfoldML` generates predictions at time-step for every patient, we take the maximum of the predictive probabilities along the prediction horizon for each patient and normalize them with a sum of 1. We compute the multi-class ROC AUC scores by averaging the pairwise ROC AUCs (known as one-vs-one) of each classes.

**Better Cost-AUC tradeoff.** We evaluate the model performance in trading off between the end-to-end AUC and spatio-temporal cost as follows: we vary the user-defined cost constraint $c$ in Eq. 2, and train `UnfoldML` repeatedly while searching the trade off in a 2D space. Each run contributes a point in the scatter plot of Figure 2b. We compute the convex hull of the searched points in the set and

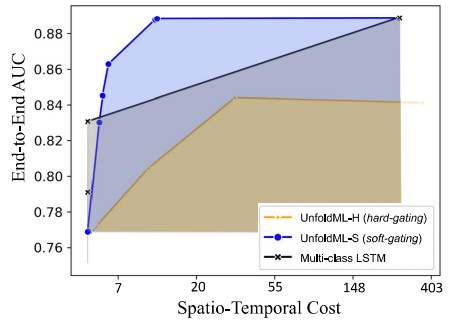 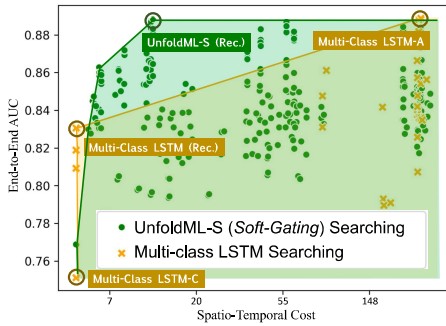

(a) Convex hull comparison for the set of points searched by different methods in the End-to-End AUC vs. Spatio-Temporal Cost tradeoff space.

(b) Search the trade off between End-to-End AUC and Spatio-Temporal cost by varying the cost constraint (x-axis is in a logarithmic scale).

Figure 2: Tradeoff space of the End-to-End AUC vs. Spatio-Temporal Cost.

| | UnfoldML-H | UnfoldML-**S** | Multi-class LSTM |
|---|---|---|---|
| Approx. area of the convex hull | 17.58 | **31.17** | 24.20 |

Table 1: The approximated area of the convex hull searched by different methods in Figure 2a.

demonstrate the resulting area of the hull in Figure 2a. We train our baseline method repeatedly by varying the LSTM architecture and plot in the same way as in Figure 2a. In comparison, `UnfoldML-S` (*soft-gating*) searches significantly higher AUC regions than the multi-class LSTM baseline and performs more consistently than the *hard-gating* algorithm. To quantify the trade-off evaluation in Figure 2a, we calculate the area of the convex hull searched by each method (Table 1).

`UnfoldML-S` presents the highest score comparing to the baseline and `UnfoldML-H`. In addition, we pick several critical points from the searched convex hull in Figure 2b and present them in Table 2. A recommended configuration for a system should effectively trade off the spatio-temporal cost and end-to-end AUC, which is to select the model along the frontier line drawn through the points that reaches the top-left corner of the hull in Figure 2b. As shown in Table 2, the recommended configuration by our pipeline, namely `UnfoldML`-S (Rec.), can achieve comparable end-to-end AUC score (88.9%) with the most accurate baseline Multi-Class LSTM-A (88.8%), while operating at a 19.6X spatio-temporal cost reduction. In comparison with `UnfoldML`-A, which naively uses the most accurate single-stage models in each stage, `UnfoldML`-S outperforms the end-to-end AUC while providing a 22.7X reduction in spatio-temporal cost. Alternatively, `UnfoldML`-S (Rec.) outperforms Multi-class LSTM (Rec.) at an AUC gain of 5.3% with only 1.3X more cost. The improvement in end-to-end AUC scores from `UnfoldML` is a result of the dynamic nature of inference pipeline, which selects the optimal pathway for each patient. The savings in spatio-temporal cost can be attributed to the usage of low cost models when they are confident enough, in contrast, baselines (`UnfoldML`-A, Multi-Class LSTM-A) always use the most accurate model.

**Single-Stage Performance.** We report single-stage performance on Sepsis and Septic Shock predictions in Table 3. For both predictions, `UnfoldML` achieves a significant reduction in costs with only a marginal loss of AUC compared to the highest accuracy baseline LSTM-A — 32.3x lower spatio-temporal cost with 1.7% lower AUC for Sepsis prediction and 26.8x lower costs with 0.7% lower AUC for Septic Shock prediction. The baseline IDK-Cascade can achieve comparable AUC as `UnfoldML`-S, but requires 3.4x higher costs.

**Better Early-hour Prediction.** Table 2 also reports the average early prediction, showing `UnfoldML`-S predicts septic shock earlier than all other methods. It predicts 2.1 hrs prior to the strongest baseline Mutli-class LSTM-A. It benefits from the multi-stage cascaded prediction in `UnfoldML` such that the system can exit early from the sepsis stage once it is in the $ICK_1$ class and proceed to shock prediction before sepsis is truly diagnosed.

### 4.2 Task 2: Subcategory Classification

Our second experiment evaluates `UnfoldML` on a computer vision task for detecting a *subcategory-of-interest*. In this *subcategory-of-interest* task, we aim to accurately predict if an image falls within a

| | Sepsis-Septic Shock Prediction | | |
| --- | --- | --- | --- |
| | end-to-end AUC (%) | Spatio-Temporal Cost Per Inference Call ($) | Early Prediction on Septic Shock (hr) |
| Multi-class LSTM-C | 75.1 | **5.1** | 12.6 |
| Multi-class LSTM-A | **88.9** | 269.0 | 24.0 |
| Multi-class LSTM (Rec.) | 83.5 | 6.0 | 15.4 |
| UnfoldML-C | 76.9 | 5.8 | 22.9 |
| UnfoldML-A | 87.2 | 311.8 | 16.4 |
| UnfoldML-H (Rec.) | 84.4 | 34.2 | 14.2 |
| UnfoldML-S (Rec.) | 88.8 | 13.7 | **26.1** |

Table 2: End-to-end performance comparison on two-stage prediction ('-C' denotes the model choice of *the cheapest and least accurate*; '-A' denotes the model choice of *the costliest and most accurate*; '-Rec.' denotes the recommended model choice with a good *trade-off between cost and accuracy* in Figure 2b; '-S' denotes *soft-gating*; '-H' denotes *hard-gating*.

| | Sepsis Prediction | | Septic Shock Prediction | |
| --- | --- | --- | --- | --- |
| | Single-Stage AUC (%) | Spatio-Temporal Cost Per Inference Call ($) | Single-Stage AUC (%) | Spatio-Temporal Cost Per Inference Call ($) |
| TREWScore (Henry et al., 2015) | - | - | 83.0 | **2.3** |
| Logistic Regression | 74.5 | **2.3** | 87.0 | **2.3** |
| Decision Tree | 70.4 | 2.4 | 83.1 | 2.4 |
| Random Forest | 75.7 | 148 | 85.2 | 148 |
| LSTM-C | 88.6 | 5.1 | 90.3 | 5.1 |
| LSTM-A | **92.8** | 268 | **96.9** | 268 |
| IDK-Cascade (Wang et al., 2017) | 91.7 | 28.1 | 95.2 | 34.0 |
| UnfoldML-S | *91.1* | *8.3* | *96.2* | *10.0* |

Table 3: Single-stage performance

specific subcategory of a dataset with classes that form a hierarchical structure, while still ensuring our spatio-temporal costs remain within the user-defined budget. For our experiment using the CIFAR-100 dataset (Krizhevsky et al., 2009), we define two separate *subcategory-of-interest* tasks based on the real-world system applications of computer vision systems. We want to identify images of a specific subcategory from one of two chosen categories: 'people' (baby, boy, girl, man, woman) and 'vehicles' (bicycle, bus, motorcycle, pickup truck, train). CIFAR-100 consists of 60,000 32x32 colour images in total of 100 'fine' classes. There are 500 training images and 100 testing images per class. In addition, each image also is assigned a 'coarse' label indicating which category (such as people, vehicles, trees, etc) it belongs to. In our experiment, we do a multi-class classification on the 'coarse-granularity' labels in Stage-1 to detect the category, then we do multi-class classification on the original 'fine-granularity' CIFAR-100 labels in the second step to identify the *subcategory-of-interest*.

### 4.2.1 Experimental Setup

Our model zoo is made of WideResNets (WRN) using Sharpness-Aware Minimization (Foret et al., 2020) which is cited as having state-of-the-art performance on the CIFAR-100 image recognition task. For each stage we train models with widths from $[2, 4, 6, 8, 10]$ and depths from $[16, 22, 24, 28]$. For Stage-1 we train $K_1$ number of multi-class classification models using the full dataset of CIFAR-100 with coarse-granularity labels as our target subcategories, resulting in a total of 20 classes. We train models for Stage-1 using the full 32x32 image and as well as random crops of size 24x24 or 16x16. For Stage-2 we train $K_2$ multi-class models to identify the next subcategory that *happens-after* the coarse subcategory from Stage-1. We train our Stage-2 models only using data with the coarse-granularity label for our subcategory in Stage-1. We train Stage-2 models using the full 32x32 image or random crop of 24x24. We profile each model by computing the number of Multiply-Accumulate Operations (MACs) and use it as proxy measure of our spatio-temporal cost.

### 4.2.2 Evaluation and Results

We compare UnfoldML against a multi-class WRN classifier baseline which directly predicts labels at the fine-granularity. We train the baseline with a width factor of 10 and a depth of 28 using the full-sized 32x32 image. We evaluate both accuracy and MACs, and report the results in Table 4. The results show that UnfoldML is able to find the optimal gating policy for the pipeline and achieve a spatio-temporal cost-savings of 6.9X with 1% reduction in accuracy for 'people' subcategory classification; we achieve a cost-savings of 4.7X with 0.4% reduction in accuracy for 'vehicle'

| | Subcategory 'People' | | Subcategory 'Vehicles' | |
|---|---|---|---|---|
| | Accuracy (%) | Macs (10M) | Accuracy (%) | Macs (10M) |
| WRN-A | **98.1** | 525.0 | **99.3** | 667.0 |
| `UnfoldML-S` | 97.1 | **77.0** | 98.9 | **142.3** |

Table 4: Performance on subcategory classification

*subcategory-of-interest* identification. Thus, `UnfoldML` achieves a savings in spatio-temporal cost for this task without compromising the overall accuracy. `UnfoldML` can accomplish this because when it is confident that queries do not belong to our *subcategory-of-interest* ($ICK_0$), we are able to *early exit* the query from the prediction pipeline. When `UnfoldML` is not confident ($IDK$), it *upgrades* queries to costlier and more accurate models. Only when `UnfoldML` can confidently predict the subcategory ($ICK_1$) in Stage-1 will it *transition* the query to Stage-2 and make the final prediction. This results in a significant cost-savings without compromising accuracy.

A recent alternative method for trading-off accuracy and cost in image classification is CwCF (Classification with Costly Features) (Janisch et al., 2019), which traverses the trade-off space by varying a cost parameter that limits the number of selected features and uses Deep Q-Learning (DQL) model to train a feature-cost-aware classifier. In addition, it also includes a pre-trained cost-unaware High-Performance Classifier (HPC), which is called when it decides to include all the features. In order to establish a fair baseline against `UnfoldML`, we used the most accurate multi-class WRN for the HPC in CwCF. We vary their cost parameter lambda for searching the optimal trade-off point in the accuracy-cost space, however, it only returns two extreme data points that gives either low accuracy of 1% with only < 10 features or high accuracy as the WRN-A shows in Table 4 with all the features. We observe similar failure in traversing the trade-off space for CIFAR-10 in Figure 4(c) of their paper, so we do not include this as a baseline for this task.

# 5   Conclusion

ML models, including for healthcare applications, are growing exponentially in size and cost of inference. This is problematic for resource constrained hospital environments. Costlier, monolithic models require expensive hardware, and can not fit on bed-side compute or even on site compute clusters with clinical implications. `UnfoldML` proposes a set of mechanisms and policies to address the growing cost of monolithic classifiers for healthcare applications. It implements a query propagation mechanism that "unfolds" a monolithic multi-class classifier into a sequence of single-class classifiers, each with its own cascade progressively more complex models. Each query is allowed to (1) confidently exit the pipeline with an $ICK_0$, (2) transition horizontally to the next stage in the pipeline with an $ICK_1$, or (3) upgrade vertically to a more complex model within the horizontal stage with an $IDK$. This mechanism is coupled with a set of policies, such as soft-gating, that set the thresholds at which the state transitions occur for queries to the system. `UnfoldML` builds on a fundamental insight that classes may have a "happens before" relationship between them, which can be leveraged to "unfold" a classifier, leading to savings in spatio-temporal cost (how much resource used for how long) and clinically significant earlier onset prediction. `UnfoldML` improves the frontier of optimality in the cost/accuracy tradeoff space and is able to nearly match (within 0.1%) SoTA AUC performance for septic shock prediction at the $\frac{1}{20}^{th}$ of the baseline cost. `UnfoldML` and the application of the "happens before" insight generalizes to computer vision tasks with 5x cost savings gained for a mere 0.4% drop in accuracy. Limitations of this work include demonstrations on more than 2 stages tasks.

# Acknowledgements

This material is based upon work supported by the National Science Foundation under Grant Numbers NSF IIS-2106961, CAREER IIS-2144338, and CCF-2029004. We would also like to acknowledge Dr. Kevin Maher and Dr. Alaa Aljiffry of Children's Healthcare of Atlanta for their medical insights and clinical guidance as well as the Neurips'22 Area Chairs and reviewers for their insightful feedback, which contributed to the improved quality of this paper. **Disclaimer:**  Any opinions, findings, and conclusions or recommendations expressed in this material are those of the authors and do not necessarily reflect the views of the National Science Foundation.

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
