# Appendix

## 5.1 Training Algorithms

**Algorithm** 1 presents the Hard-Gating Training Algorithm, **Algorithm** 2 presents the Soft-Gating Training Algorithm, and **Algorithm** 3 summarizes the End-to-End Training Algorithm.

---

**Algorithm 1** *Hard-gating* Algorithm for In-Stage $IDK$ Cascade

---

**Input**
 $\mathcal{D}^s$: Training data containing $N^s$ samples in stage-$s$
 $\mathcal{M}^s$: Sorted list of the models trained for stage-$s$
 $\mathcal{C}$: Dictionary of models' spatio-temporal costs
 $c_s$: User-defined budget of spatio-temporal cost for stage-$s$
 $q$: Confidence function
 $maxA$: Value for the upper bound of the cutoffs to avoid over-fitting
 $nBins$: Number of bins for the grid search
**Output**
 $\boldsymbol{\alpha}_s^*$: The optimal IDK cutoff vector for stage-$s$

1: **procedure** HARDGATING($\mathcal{D}^s, \mathcal{M}^s, c_s, \mathcal{C}, q, maxA, nBins$)
2:      $\boldsymbol{\alpha}_s^* = []$, $ModelAssign = \mathbf{1}$, $cost = \sum_{i,t} \mathcal{C}[m_{s1}]$
3:      **if** $cost > c_s$ **then return** $\boldsymbol{\alpha}_s^*$
4:      **end if**
5:      **for** $k$ in range($K_s - 1$) **do**                ▷ Bottom-up search
6:          $Idx4k \leftarrow \cup I(ModelAssign[i,t] == k)$.
7:          **if** $Idx4k$ is $\emptyset$ **then break**
8:          **end if**
9:          $minQ \leftarrow \min_{Idx4k} \{q_{sk}(\boldsymbol{x}_{it})\}$
10:         $maxQ \leftarrow \min(maxA, \max_{Idx4k} \{q_{sk}(\boldsymbol{x}_{it})\})$.
11:         $\alpha_{sk}^* \leftarrow minQ$.
12:         **for** $\alpha_{sk}$ in $LinSpace(minQ, maxQ, nBins)$ **do**
13:             $IDK \leftarrow \cup_{Idx4k} I(q_{sk}(\boldsymbol{x}_{it}) \in [\alpha_{sk}^*, \alpha_{sk}))$
14:             **if** $IDK$ is not $\emptyset$ **then**
15:                 **if** $cost + \sum_{IDK} \mathcal{C}[m_{sk+1}] - \mathcal{C}[m_k] > c_s$ **then**
16:                     $\hat{\boldsymbol{\alpha}}_s \leftarrow \boldsymbol{\alpha}_s^* + [\alpha_{sk}^*]$; **return** $\boldsymbol{\alpha}_s^*$
17:                 **end if**
18:                 $\alpha_{sk}^* \leftarrow \alpha_{sk}$,
19:                 $ModelAssign[IDK] \leftarrow k + 1$,
20:                 $cost+ = \sum_{IDK} \mathcal{C}[m_{sk+1}] - \mathcal{C}[m_k]$
21:             **end if**
22:         **end for**
23:         $\boldsymbol{\alpha}_s^* \leftarrow \boldsymbol{\alpha}_s^* + [\alpha_{sk}^*]$
24:      **end for**
25:      **return** $\boldsymbol{\alpha}_s^*$
26: **end procedure**

---

## 5.2 Data preparation

By following the definition of Sepsis-3 Singer et al. (2016), we identify the sepsis onset to be the time when an increase in the Sequential Organ Failure Assessment (SOFA) score of 2 points or more occurs in response to infections. We use the *Sepsis-3* toolkit[3] to obtain the suspected infection time in patients, and following the process in Seymour et al. (2016) to finally label the onset of sepsis. We result at a total number of $20,009$ sepsis patients out of the $52,902$ adult patients from MIMIC-III database. We exclude those patients who stay in ICUs less than 6 hours and also exclude those patients who developed sepsis within the first 6 hours after ICU admission. This reduces our cohort to a total of $34,475$ ICU patient, and only $2,370(6.8\%)$ out of them are labeled as sepsis (because $88.1\%$ of sepsis onsets happened within the first 6 hours after ICU admission and are excluded from our study cohort). Then according to Singer et al. (2016), we identify the onset of septic shock as

---

[3]https://doi.org/10.5281/zenodo.1256723

---

**Algorithm 2** *Soft-gating* Algorithm for In-Stage $IDK$ Cascade

---

**Input**

   $\mathcal{D}^s$: Training data containing $N^s$ samples in stage-$s$

   $\mathcal{M}^s$: Sorted list of the models trained for stage-$s$

   $\boldsymbol{f}_{\mathcal{M}^s}$: A multi-head DKD model for distilling all the model's confidence at stage-$s$

   $\mathcal{C}$: Dictionary of models' spatio-temporal costs

   $c_s$: User-defined budget of spatio-temporal cost for stage-$s$

   $q$: Confidence function

   $\lambda$: Controller for the spatio-temporal cost budget

   $\mu$: Controller for L1-norm sparsity regularization

   $nEpochs$: Number of training epochs

**Output**

   $\boldsymbol{a}_s^*, \boldsymbol{b}_s^*$: the optimal soft-gating IDK coefficient for stage-s

 1: **procedure** SOFTGATING($\mathcal{D}^s, \mathcal{M}^s, \boldsymbol{f}_{\mathcal{M}^s}, c_s, \mathcal{C}, q$)
 2:    $lr \leftarrow 1e-1, e \leftarrow 0, \boldsymbol{a}_s \leftarrow 1, \boldsymbol{b}_s \leftarrow 0.5$
 3:    **while** e < nEpochs **do**
 4:       $\hat{q}_{sj}(\boldsymbol{x}_t) \leftarrow q\big(\boldsymbol{f}(\boldsymbol{x}_t; m_{sk})[1]/\sum \boldsymbol{f}(\boldsymbol{x}_t; m_{sk})\big)$         ▷ DKD confidence distillation
 5:       $\mathcal{L}_{\text{sparse}}^s \leftarrow \sum_{i,t,k} \| G^{IDK}(\hat{q}_{sj}(\boldsymbol{x}_t)) \|_1$
 6:       $\mathcal{L}_{\boldsymbol{a}_s, \boldsymbol{b}_s} \leftarrow \mathcal{L}_{\text{nll}}^s + \lambda \mathcal{L}_{\text{cost}}^s + \mu \mathcal{L}_{\text{sparse}}^s$
 7:       Optimize $\mathcal{L}_{\boldsymbol{a}_s, \boldsymbol{b}_s}$ using SGD
 8:       Reduce $lr$ by factor $0.5$ once learning stagnates.
 9:       $e \leftarrow e + 1$
10:    **end while**
11:    **return** $\boldsymbol{a}_s^*, \boldsymbol{b}_s^*$
12: **end procedure**

---

---

**Algorithm 3** End-to-End Training algorithm for `UnfoldML`

---

**Input**

   $\mathcal{D}$: Full training data containing $N$ instances

   $\mathcal{M}$: Full model zoo

   $\mathcal{C}$: Dictionary of models' spatio-temporal costs

   $q$: Confidence criterion

**Output**

   $\boldsymbol{\theta}^*$: the optimal ICK$_1$ gate parameters

   $\boldsymbol{\alpha}^*$ (or $\boldsymbol{a}^*, \boldsymbol{b}^*$): the optimal IDK gate parameters

 1: **procedure** END-TO-ENDTRAINING($\mathcal{D}, \mathcal{M}$)
 2:    Pre-allocate costs $c_s$ for each stage s.
 3:    **Step 1:** Learn in-stage IDK gate parameters.
 4:    **for** each stage s **do**
 5:       $\boldsymbol{\alpha}^* \leftarrow$ HardGating($\mathcal{D}^s, \mathcal{M}^s, c_s, C, q$)
 6:       or, $\boldsymbol{a}^*, \boldsymbol{b}^* \leftarrow$ SoftGating($\mathcal{D}^s, \mathcal{M}^s, c_s, C, q$)
 7:    **end for**
 8:    ______________________________________
 9:    **Step 2:** Learn ICK$_1$ gate parameters.
10:    **for** each model $m_{sk}$ **do**
11:       $\theta_{sk}^* \leftarrow$ Grid Search for minimizing $\sqrt{\epsilon_{\text{ICK}_1}^2 + \epsilon_{\text{ICK}_0}^2}$
12:    **end for**
13:    **return** $\boldsymbol{\alpha}^*$ (or $\boldsymbol{a}^*, \boldsymbol{b}^*$), $\boldsymbol{\theta}^*$
14: **end procedure**

---

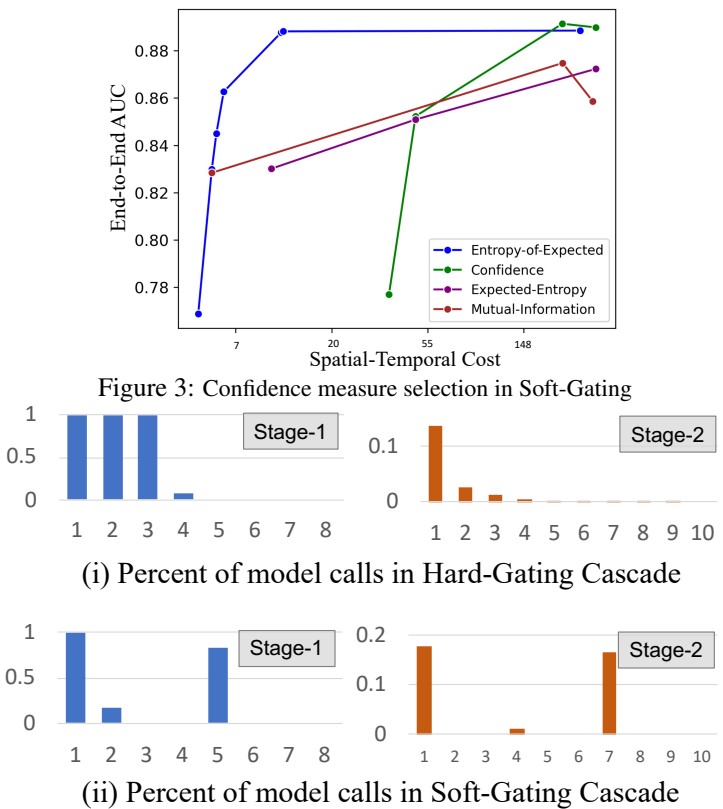

Figure 3: Confidence measure selection in Soft-Gating

(i) Percent of model calls in Hard-Gating Cascade

(ii) Percent of model calls in Soft-Gating Cascade

Figure 4: Transitions in model calls: both cascades always call the first model per each stage for an entrance and transition to next models (IDK) or next stage (ICK).

when a vasopressor is required to maintain a mean arterial pressure (MAP) $\geq 65$ mm Hg and serum lactate level $> 2$ mmol/L ($> 18$ mg/dL). We result at $229(9.7\%)$ septic shock patients out of the $2,370$ sepsis patients.

For feature generation, we extract 8 patient static characteristics including age, gender, race, height, weight, sepsis onset hour since ICU admission, whether diagnosed diabetes or on a ventilator at ICU admission. Then we extract the dynamic features by obtaining the 8 vital signs, 16 lab measurements, 6 vassopressors, continuous replacement therapies (CRRT), ventilation, 2 intravenous fluids in fluid resuscitation, and 5 additional measurements that are recommended for monitoring during sepsis management. The 8 vital signs include heart rate, systolic blood pressure, diastolic blood pressure, mean blood pressure, respiration rate, temperature, SpO2 and glucose. The 16 lab measurements include Anion gap, Albumin, Bands, Bicarbonate, Bilirubin, Creatinine, Chloride, Glucose, Hematocrit, Hemoglobin, Lactate, Platelet, Potassium, PTT, INR, PT, Sodium, BUN and WBC. The 6 vasopressors include dobutamine, dopamine, epinephrine, norepinephrine, phenylephrine, and vasopressin. The 2 fluids include Crystalloids and Colloids that are recommended in the early management of sepsis Rhodes et al. (2017), and particularly fluid resuscitation of bolus $\geq 500$ mL is one of the most common treatment for managing septic shock. The 5 additional measurements include whether a vasopressor is needed to maintain a mean arterial pressure (MAP) $\geq 65$ mm Hg, serum lactate level $> 2$ mmol/L, urine output $\geq 5$ ml/kg/hr, venous oxygen saturation (SvO2) $\geq 70\%$ and central venous pressure (CVP) of $8 - 12$ mmHg. We fill missing values like lab measurements using the last measured value; we clamp real-valued features in between their $0.05$-quantile and $0.95$-quantile values respectively and normalize the features using min-max normalization.

For training sepsis prediction models, we take the full training cohort but discard the data after the first sepsis onset in sepsis patients, then we label the data per hour, and label the current sepsis outcome as 1 if the true sepsis is going to happen in the next 12 hours (designed for early prediction on sepsis). For training shock prediction models, we take the sepsis sub training cohort and discard the data before sepsis onset. We also discard the data after septic shock onset in shock patients. Then we label in the same way as for sepsis, i.e. label the current shock outcome as 1 if the true shock will take place in the next 12 hrs. For those non sepsis patients, we discard the first 12 hrs data after ICU

| Model | Model Zoo | | | | Dirichlet Knowledge Distillation (DKD) | | | | |
|---|---|---|---|---|---|---|---|---|---|
| | AUC | Computational Cost | Data Modality | Total Norm. Cost | AUC | MAE Confidence | MAE Entropy of Exp. | MAE Entropy | MAE MI |
| vitals_1hr.h100.nlayer1 | 74.5% | 5 | 1 | 0.10 | 74.4% | 0.07 | 0.13 | 0.08 | 0.05 |
| vitals_6hr.h100.nlayer1 | 78.2% | 7 | 1 | 0.11 | 74.7% | 0.06 | 0.11 | 0.07 | 0.05 |
| vitals_6hr.h100.nlayer3 | 79.7% | 172 | 1 | 0.54 | 74.8% | 0.08 | 0.14 | 0.09 | 0.06 |
| vitals_6hr.h300.nlayer2 | 81.1% | 173 | 1 | 0.54 | 75.0% | 0.08 | 0.14 | 0.09 | 0.06 |
| vitals_12hr.h200.nlayer4 | 82.3% | 175 | 1 | 0.55 | 70.5% | 0.09 | 0.16 | 0.10 | 0.10 |
| vitals_labs_1hr.h100.nlayer1 | 76.8% | 5 | 2 | 0.20 | 74.0% | 0.06 | 0.11 | 0.07 | 0.04 |
| vitals_labs_6hr.h100.nlayer1 | 81.8% | 86 | 2 | 0.41 | 74.0% | 0.06 | 0.12 | 0.08 | 0.04 |
| vitals_labs_6hr.h100.nlayer2 | 82.6% | 257 | 2 | 0.86 | 73.5% | 0.06 | 0.12 | 0.08 | 0.05 |
| vitals_labs_6hr.h100.nlayer3 | 82.5% | 258 | 2 | 0.86 | 74.2% | 0.08 | 0.14 | 0.09 | 0.05 |
| vitals_labs_csu_1hr.h100.nlayer1 | 78.3% | 5 | 3 | 0.30 | 73.8% | 0.07 | 0.13 | 0.09 | 0.05 |
| vitals_labs_csu_6hr.h100.nlayer1 | 81.6% | 90 | 3 | 0.52 | 73.4% | 0.08 | 0.14 | 0.10 | 0.05 |
| vitals_labs_csu_6hr.h400.nlayer1 | 81.6% | 258 | 3 | 0.96 | 73.5% | 0.05 | 0.11 | 0.07 | 0.05 |
| vitals_labs_csu_6hr.h400.nlayer3 | 83.5% | 259 | 3 | 0.97 | 73.7% | 0.07 | 0.13 | 0.08 | 0.06 |
| vitals_labs_csu_6hr.h300.nlayer3 | 82.2% | 264 | 3 | 0.98 | 73.5% | 0.07 | 0.14 | 0.08 | 0.07 |
| vitals_labs_csu_6hr.h100.nlayer2 | 81.8% | 272 | 3 | 1.00 | 73.2% | 0.09 | 0.15 | 0.10 | 0.06 |
| vitals_labs_csu_12hr.h300.nlayer4 | 85.1% | 268 | 3 | 0.99 | 72.6% | 0.09 | 0.16 | 0.10 | 0.07 |

Table 5: Sepsis-Stage model zoo

| Model | Model Zoo | | | | Dirichlet Knowledge Distillation (DKD) | | | | |
|---|---|---|---|---|---|---|---|---|---|
| | AUC | Computational Cost | Data Modality | Total Norm. Cost | AUC | MAE Confidence | MAE Entropy of Exp. | MAE Entropy | MAE MI |
| vitals_1hr.h100.nlayer1 | 87.0% | 5 | 1 | 0.23 | 87.1% | 0.05 | 0.07 | 0.04 | 0.03 |
| vitals_6hr.h100.nlayer1 | 88.6% | 7 | 1 | 0.23 | 86.7% | 0.04 | 0.08 | 0.06 | 0.03 |
| vitals_6hr.h100.nlayer3 | 88.4% | 172 | 1 | 0.28 | 86.9% | 0.04 | 0.08 | 0.06 | 0.03 |
| vitals_6hr.h300.nlayer2 | 86.8% | 173 | 1 | 0.28 | 86.5% | 0.04 | 0.07 | 0.06 | 0.03 |
| vitals_12hr.h300.nlayer2 | 88.6% | 174 | 1 | 0.29 | 86.0% | 0.04 | 0.09 | 0.07 | 0.03 |
| vitals_12hr.h200.nlayer4 | 88.6% | 175 | 1 | 0.29 | 85.3% | 0.04 | 0.09 | 0.06 | 0.04 |
| vitals_12hr.h300.nlayer3 | 85.1% | 177 | 1 | 0.29 | 85.3% | 0.05 | 0.11 | 0.08 | 0.04 |
| vitals_12hr.h400.nlayer3 | 89.5% | 189 | 1 | 0.29 | 85.6% | 0.03 | 0.07 | 0.05 | 0.02 |
| vitals_labs_1hr.h100.nlayer1 | 89.0% | 5 | 2 | 0.45 | 85.4% | 0.03 | 0.06 | 0.04 | 0.03 |
| vitals_labs_6hr.h100.nlayer1 | 89.8% | 86 | 2 | 0.48 | 86.1% | 0.04 | 0.08 | 0.05 | 0.04 |
| vitals_labs_6hr.h100.nlayer2 | 89.9% | 257 | 2 | 0.54 | 85.4% | 0.03 | 0.06 | 0.04 | 0.04 |
| vitals_labs_6hr.h300.nlayer1 | 87.7% | 258 | 2 | 0.54 | 84.0% | 0.03 | 0.07 | 0.04 | 0.04 |
| vitals_labs_6hr.h200.nlayer2 | 89.8% | 263 | 2 | 0.54 | 87.4% | 0.04 | 0.09 | 0.06 | 0.04 |
| vitals_labs_12hr.h300.nlayer4 | 93.5% | 262 | 2 | 0.54 | 82.9% | 0.01 | 0.03 | 0.02 | 0.01 |
| vitals_labs_12hr.h200.nlayer4 | 90.7% | 270 | 2 | 0.54 | 89.1% | 0.01 | 0.04 | 0.02 | 0.02 |
| vitals_labs_csu_1hr.h100.nlayer1 | 90.8% | 5 | 3 | 0.68 | 86.2% | 0.04 | 0.09 | 0.06 | 0.04 |
| vitals_labs_csu_6hr.h100.nlayer1 | 91.9% | 90 | 3 | 0.71 | 86.3% | 0.04 | 0.10 | 0.06 | 0.05 |
| vitals_labs_csu_1hr.h100.nlayer4 | 90.2% | 172 | 3 | 0.73 | 86.7% | 0.02 | 0.05 | 0.04 | 0.02 |
| vitals_labs_csu_6hr.h200.nlayer2 | 88.9% | 258 | 3 | 0.76 | 86.3% | 0.02 | 0.06 | 0.04 | 0.03 |
| vitals_labs_csu_6hr.h300.nlayer3 | 88.7% | 264 | 3 | 0.77 | 88.0% | 0.01 | 0.02 | 0.01 | 0.01 |
| vitals_labs_csu_12hr.h200.nlayer3 | 92.1% | 287 | 3 | 0.78 | 86.2% | 0.02 | 0.05 | 0.03 | 0.02 |
| vitals_labs_csu_med_1hr.h100.nlayer1 | 91.5% | 5 | 4 | 0.90 | 85.5% | 0.03 | 0.06 | 0.03 | 0.05 |
| vitals_labs_csu_med_6hr.h100.nlayer1 | 91.6% | 86 | 4 | 0.93 | 87.3% | 0.03 | 0.07 | 0.04 | 0.04 |
| vitals_labs_csu_med_6hr.h400.nlayer1 | 91.4% | 257 | 4 | 0.99 | 87.1% | 0.03 | 0.08 | 0.05 | 0.04 |
| vitals_labs_csu_med_6hr.h300.nlayer3 | 90.4% | 259 | 4 | 0.99 | 86.4% | 0.03 | 0.07 | 0.05 | 0.03 |
| vitals_labs_csu_med_12hr.h100.nlayer2 | 93.4% | 262 | 4 | 0.99 | 83.3% | 0.00 | 0.01 | 0.01 | 0.01 |
| vitals_labs_csu_med_12hr.h400.nlayer4 | 93.4% | 269 | 4 | 0.99 | 84.7% | 0.02 | 0.06 | 0.02 | 0.04 |

Table 6: Septic Shock-Stage model zoo

admission to reduce data noises and randomly sample a sequence length between 12 hrs up to 7 days per each non sepsis patients. More details of data prepossessing are provided in the attached code.

## 5.3 Model Zoo

Computational cost was measured in $ms$ as the total running time of feeding all the test data (with batch size of 256) calling each individual models on a single `GeForce RTX 2080Ti` divided by the total number of calls. Then we multiply the cost by 10 as the GPU is approximately 10X hardware cost comparing to a CPU. Future work can extend the model zoo to include CPU models or running all the models on CPUs based on resource specifications. Table 5 and Table 6 respectively show the model prediction AUC scores on the validation set for the sepsis and septic shock stages.

In addition, we also fit small DKD surrogate models for distilling the predictive probabilities and confidence of the models in the zoo. The DKD model is a 4-layer MLP taking the embedding vectors from the first model in each stage, so it obtains similar AUC scores on the validation set comparing to the early models in the zoo but much lower scores comparing to the later heavier models. But the mean absolute errors (MAE) of the DKD model on estimating confidence measures of the original models are consistently small, which is beneficial for our soft-gating algorithm that requires only confidence estimation instead of predictive probabilities.

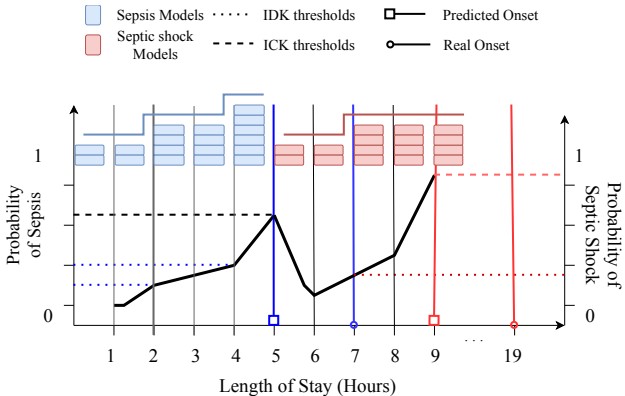

Figure 5: A timeline shown for an example shock patient. The y-axis represents probabilities of sepsis (left) and septic shock (right). The x-axis represents patient's length of stay (hours). This figure illustrates how different models are selected based on patient's critical health condition and timely septic shock prediction is made in cost-efficient manner.

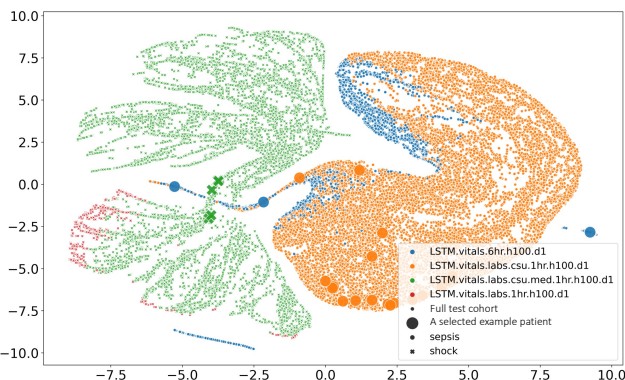

Figure 6: Dynamic model allocations in the `UnfoldML`: the example shock patient (in large-sized marker) transitioned from cheaper model in sepsis (dot) stage to costlier model in shock stage (cross).

### 5.4 Model Utilization in `UnfoldML`.

We analyze model utilization frequency (the proportion of how many times a model was invoked) in our test cohort and compare model frequencies for hard and soft gating in Figure 4 (models are grouped into 8 groups for Stage 1 and 10 groups for Stage 2): soft gating can skip invocations of many models and directly select the more confident models for faster transitioning to shock stage.

where $\psi$ is the *digamma* function defined as the logarithmic derivative of the gamma function, $\alpha_0$ and $\alpha_1$ are the concentration parameters estimated by the DKD models. More definitions are in Malinin and Gales (2018). We show "Entropy of expected" exhibits the best AUC-Cost trade-off path in Figure 3.

### 5.5 Qualitative Evaluation

We walk through an example shock patient's length of stay in ICU from the test set, and deploy the proposed multi-stage prediction pipeline on it. `UnfoldML` starts the prediction of sepsis with a cheaper model as seen in Figure 5. At t=2, the model's prediction probability reaches the $IDK$ threshold which signifies model's uncertainty in sepsis prediction. Hence, the `UnfoldML` switches to a costlier and more accurate model (a similar trend is observed at t=4,7). At t=5, `UnfoldML` predicts sepsis onset as the probability of sepsis prediction reaches $ICK$ threshold. Note, once sepsis onset is predicted by the cascade, it switches to a cheaper model which predicts septic shock (Stage-2). Due to early switching, `UnfoldML` can detect septic shock significantly earlier. In Stage-2, `UnfoldML` transitions to costlier model once the cheaper model becomes uncertain. Lastly, it predicts septic shock once the probability of septic shock detection reaches the $ICK_1$ threshold.

Additionally, we randomly slice 35k time-steps from the sequential data in the test set and visualize them in a TSNE Van der Maaten and Hinton (2008) plot in Figure 6 based on their embedding vectors

generated from the LSTMs in the model zoo. Different colors show the different model allocations for the subsampled test data points, sepsis (dot) and shock (cross) stages are clearly separated in to the left and right regions of the 2-D transformation space. We highlight the picked shock patient (with significantly large markers) showing its dynamic model allocations and stage transitions within `UnfoldML`.