# OpenReview forum: "UnfoldML: Cost-Aware and Uncertainty-Based Dynamic 2D Prediction for Multi-Stage Classification"
_NeurIPS.cc/2022/Conference — NeurIPS 2022 Accept_

### Official Review · Reviewer_5CL8 · 2022-07-10

**Rating:** 6
**Confidence:** 2
**Soundness:** 3 good
**Presentation:** 3 good
**Contribution:** 3 good

**Summary:**

This paper proposed a cost-aware prediction pipeline that "unfolds" a single multi-class classifier into a series of single-stage classifiers, reducing its deployment cost. To be more specific, three classes of gate functions are learned to maximize the system-wise accuracy while minimizing the prediction cost.

**Questions:**

The method focuses on the "subcategory-of-interest" task, would it be generalized to classification on all classes?


**Limitations:**

The author discussed the limitation but not negative societal impacts of the work.

**Strengths And Weaknesses:**

Strengths
  - the spatial-temporal cost of inference is reduced by orders of magnitude by the method.
  - the proposed method navigates a more complete cost-AUC trade-off space.

Weakness
  - The experiments focus mainly on two-stage tasks (such as cifar100, first on coarse classes then fine classes), it would be interesting to demonstrate the effectiveness on more than 2 stage tasks.
  - Some explanations with a simple example (such as classifying CIFAR-100) to illustrate the method would be easier to understand.
  - The method focuses on the "subcategory-of-interest" task, would it be generalized to classification on all classes?

---

> ### Author Response · Authors · 2022-08-02
> **Thanks for your valuable comments**
>
> Thank you for your valuable comments. We address your questions as follows:
>
> 1. *“The method focuses on the "subcategory-of-interest" task, would it be generalized to classification on all classes?”*
>
> UnfoldML won’t present advantages in cost reduction if it targets classification on all classes. Focusing on subcategory classification allows most of the samples that belong to ICK_0 class (i.e., subcategories likely to be not of interest) exit the pipeline early at stage 1, only the small portion of samples that belong to ICK_1 class (subcategories likely to be of interest) will enter stage 2 classification. This is the key to how UnfoldML saves inference cost compared to the baseline. In contrast, classification on all classes won’t allow early exit of the pipeline at stage 1, but it needs to propagate all the queries to different branches in stage 2 for classifications on all fine classes. So this will fail in inference cost reduction.
>
> 2. *Limitations on negative societal impacts of the work.*
>
> Actually the checklist says No to this because we in fact expect a positive societal impact, due to more than an order of magnitude reduction in spatio-temporal cost of ML inference. It has been shown that (1) there’s a direct correlation between ML inference cost and CO2 emissions in large datacenters and (2) the majority of datacenter compute cost comes from running pre-trained models in production (i.e., inference). This paper’s key goal is to minimize that cost (20x improvement) with minimal effect on AUC (<0.1%) .
>
> 3. *Others*
>
> We also state in Section 5 that limitations of this work include the current experiment only apply to 2-stage tasks. We’d like to conduct experiments on more than 2 stages in future work.
>
> We will add explanations on the image task in the final version for better understanding of the method. Thank you for the useful suggestion.

---

> > ### Comment · Reviewer_5CL8 · 2022-08-09
> > **Thank you for the response**
> >
> > Thanks to the authors for answering my question, I would keep my original score.

---

> > > ### Author Response · Authors · 2022-08-09
> > > **Thanks for your support and appreciation for this work!**
> > >
> > > Thanks for your support and appreciation for this work. It is very encouraging.

---

### Official Review · Reviewer_qEBg · 2022-07-11

**Rating:** 5
**Confidence:** 4
**Soundness:** 2 fair
**Presentation:** 3 good
**Contribution:** 3 good

**Summary:**

This paper aims to unfold a monolithic multi-class classifier to a series of single-stage classifiers for reducing deployment cost. The proposed method is a dynamic 2D projection pipeline which makes ICK predictions on sequential multi-stage classification tasks. The topic is interesting and this manuscript shows promising results on two real-world tasks.

**Questions:**

More detailed comments are as follows:
- What exactly "spatio-temporal cost" means?
- Would a lower-cost model interference must result in accuracy degradation?
- Two training algorithms (i.e., hard-gating and soft-gating), what is the relationship beween them? What is the method novelty here?
- Comparison methods in Tables are relatively out-of-date. Are there any newer methods for comparison on this reserch topic?
- In Sec. 4.2, it is claimed "our second experiment is a computer vision task demonstrating the generalizability of UnfoldML". The claimed generalizability is in terms of what? How is it validated?

Minor comments:
- Typo in Line 306, missing Eq. number.
- Tables should add standard deviations of the results.

**Strengths And Weaknesses:**

Strengths:
- The topic of unfolding a multi-class classifier to a series of single-stage calssifiers is interesting
- Experiment on real-world dataset with a relatively large number of samples

Weakness:
- Methodological novelty is unclear
- Lack of experimental comparison with recent methods
- There is no theoretical analysis of the proposed method

---

> ### Author Response · Authors · 2022-08-02
> **Thanks for your constructive reveiws.**
>
> We appreciate your constructive reviews and address your questions in the following list:
>
> 1. *“What exactly "spatio-temporal cost" means?”*
>
> Spatio-temporal cost is a multiplication of temporal cost (i.e. latency, the time spent by a model to generate a prediction) and spatial cost (i.e., hardware cost, cpu or gpu (10x expensive to cpu) used in our experiments). This cost mirrors the one commonly used for cloud billing. More details of spatio-temporal cost have been explained in Appendix line 625-629. Abstractly, we believe this success metric captures well the reduction in inference complexity AND the ability of the algorithm to choose cheaper cpu models — both factoring into the spatio-temporal cost.
>
> 2. *“Would a lower-cost model interference must result in accuracy degradation?”*
>
> No. This is why in the hard-gating algorithm, we need to first sort the model list in a monotonically increasing order w.r.t AUC and cost by discarding the sub-optimal models (line 185-186). But there is no need to discard any models in the soft-gating algorithm as the MoE method will learn to assign low weights to these sub-optimal models.
>
> 3. *“Two training algorithms (i.e., hard-gating and soft-gating), what is the relationship beween them? What is the method novelty here?”*
>
> Hard-gating algorithm inherits the hard thresholding setting from baseline IDK-cascade (line 296-297). The novelty of this algorithm is that we extend this existing solution, which is only designed for 1-D prediction, to 2-D predictions. But the grid searching algorithm enforced by the non-differentiable hard-thresholding objective leads to limited searching efficiency for the hard-gating algorithm. To overcome this, we reformulate the problem in the form of MoE and propose a novel soft-gating algorithm that can learn the gate parameters more efficiently using gradient descent. In more details, the constraint on inference time presents unique challenges on adapting MoE into UnfoldML, so we design DKD to distill model uncertainty instead of running each model in the soft-gating algorithm.
>
> 4. *“Comparison methods in Tables are relatively out-of-date. Are there any newer methods for comparison on this reserch topic?”*
>
> To the best of our knowledge, UnfoldML is the first system to provide 2-D cascading predictions. The most recent and relevant cascading algorithms IDK-cascade (line 296-297) and CwCF (line 373-380) are compared in the paper. The most reasonable baseline is the multi-task classifiers that ignores the happen-after relationship between the tasks (line 281-282). We choose LSTM and WRN as the backbone models for the two applications respectively, as they are still commonly used in the two areas.
>
> 5. *“In Sec. 4.2, it is claimed "our second experiment is a computer vision task demonstrating the generalizability of UnfoldML". The claimed generalizability is in terms of what? How is it validated?”*
>
> Because UnfoldML was initially motivated in clinical settings where disease progression naturally presents a ‘happen after’ relationship in stage transitioning. But we show it’s also generalizable to other prediction tasks such as image classifications where the ‘coarse-fine’ labels can be viewed as a ‘happen after’ relationship too.
>
> 6. *Minor comments*
>  We have fixed the typo in the rebuttal revision (line 302). We will update the table for the final version.

---

> ### Author Response · Authors · 2022-08-09
> **Response to Reviewer qEBg**
>
> We thank the reviewer for their insightful feedback, thorough understanding of our contributions, and great questions. In light of the upcoming deadline, we'd like to circle back and confirm with the reviewer whether our response addressed all of your concerns and if there're any additional questions we could help clarify.

---

### Official Review · Reviewer_cok9 · 2022-07-11

**Rating:** 4
**Confidence:** 5
**Soundness:** 2 fair
**Presentation:** 2 fair
**Contribution:** 2 fair

**Summary:**

The authors propose a cost-aware 2D prediction pipeline UnfoldML to solve multi-stage classification problem under the limited resources. In order to better find the optimal trade-off between model performance and cost, the author define a search space and three gate parameters (ICK_0, IDK, ICK_1) to navigate the trade-off. In addition, two learning algorithms (hard-gating traning algorithm and soft-gating training algorithm) are proposed to solve this optimization problem with search space. Finally, two experiments on two different real-world tasks show that the proposed method can achieve good performance as well as reducing the spatio-temporal cost compared to other baseline models.


**Questions:**

Does the cheaper algorithm DKD have the similar prediction with the expensive MoE algorithm on gate parameters?

How long will it take to search an optimal cost-AUC tradeoff in the defined search space?

According to Table 2, The proposed method UnfoldML-S (Rec.) performs worse than the baseline Multu-class LSTM (Rec.), because both have similar AUC but the proposed method has more cost and worse early prediction hours.

**Limitations:**

For the search sapce, the vertical space is not well-designed. According to the experiments, at each stage, the authors only train models with different predefined setting of architectures and different combination of predefined feature sets.

There is no proof, experiment or data to validate that the proposed (hard-gating or soft-gating) algorithms find the optimal policy in the defined search space. It may be sub-optimal.

The authors only did experiments on two-stage classification task, which is simple on multi-stage classification.



**Strengths And Weaknesses:**

Compared to previous work, the authors defined a cost-performance search space and propose two algorithms to solve this optimization problem. The authors did lots of experiments with different models and evaluate the model performance using AUC curve instead of accuracy (can be biased).


The proposed hard-gating training algorithm is unstable. Its performance is expected to vary with the upper bound maxA.

The format of this paper is not good. Equations and figures are too small, and there are lots of typos. For example, line 240: Hard-Gating (3); line 306: Eq. ??; line 308: figure 3; line 314: Table 4.1.2.

Didn't find Appendix section

---

> ### Author Response · Authors · 2022-08-02
> **Thanks for your valuable reviews.**
>
> We thank you for your valuable reviews. We address your questions as follows:
>
> 1. *“Does the cheaper algorithm DKD have the similar prediction with the expensive MoE algorithm on gate parameters?”*
>
> We did not have this comparison since running the expensive MoE over the full model zoo is too time consuming for the experiment. Alternatively, to understand how well the DKD model can distill the true model’s confidence scores, we did evaluate the mean absolute errors (MAEs) of their uncertainty prediction on the test set. We got an average MAE of 0.07 (sd of 0.2) for the distillation of all the models in the zoo, which demonstrates that the DKD algorithms are able to properly capture the models’ uncertainty without really running them. We add these evaluation results of the DKD models in Appendix Table 5 and 6 within the supplementary.zip in rebuttal revision.
>
> 2. *“How long will it take to search an optimal cost-AUC tradeoff in the defined search space?”*
>
> The time for searching an optimal cost-AUC tradeoff depends on the slice granularity of varying the cost constraints in its range (min and max model cost in the model zoo) at each stage. In our sepsis-shock experiment, learning gate parameters given a fixed cost constraint takes about 4 minutes (on a single NVIDIA GeForce RTX2080). It took about 16 hours to traverse the space when we evenly sliced 50 values on cost constraint per each stage.
>
> We want to clarify that searching the entire cost-AUC tradeoff space is not needed for configuring a UnfoldML prediction pipeline. A configuration of UnfoldML only corresponds to one single dot in the plot of Figure 2b: the user will first specify their budget (including hardware like cpu or gpu, latency requirement for serving model predictions e.g.,  1ms or 0.1ms, etc) as the cost constraint c. UnfoldML will then learn proper gate parameters for maximizing the system-wide inference accuracy while satisfying the specified cost budget c.
> Searching for an optimal cost-AUC tradeoff in the space makes a recommendation on what the most beneficial configuration of UnfoldML (i.e. the best balance between cost and AUC) would be, regardless of a user’s predefined budget.
>
> 3. *“According to Table 2, The proposed method UnfoldML-S (Rec.) performs worse than the baseline Multu-class LSTM (Rec.), because both have similar AUC but the proposed method has more cost and worse early prediction hours.”*
>
> This has been mis-read. In Table 2, the proposed method UnfoldML-S (Rec.) performs better than the baseline Multi-class LSTM (Rec.), since both have similar AUC (88.8 vs. 88.9) while the proposed method has ~20x lower cost (13.7 vs. 269.0) and earlier prediction (26.1 hours earlier vs. 24.0 earlier).
>
> We address the limitations as follows:
> 1. *“For the search space, the vertical space is not well-designed. According to the experiments, at each stage, the authors only train models with different predefined setting of architectures and different combination of predefined feature sets.”*
>
> The focus of our method is on how to serve the models that have already been trained  rather than train the models in the zoo. Training procedures like model architecture searching in NAS is beyond the scope of this paper.
>
> 2. *“There is no proof, experiment or data to validate that the proposed (hard-gating or soft-gating) algorithms find the optimal policy in the defined search space. It may be sub-optimal.”*
>
> This is a fair point. However, because the optimization objective is not convex,  none of the baseline methods nor the proposed method can guarantee the learnt optimal policy is the global optimum. Instead, with experiment results (e.g. Figure 2a and 2b) we aim to show that the proposed soft-gating algorithm can cover larger and explore better areas in the searching space so it is able to find better optimal policies compared to the baselines.
>
> 3, *“The authors only did experiments on two-stage classification task, which is simple on multi-stage classification.”*
>
> True. We have stated this limitation in our paper in Section 5. We believe that arbitrary DAG pipeline support is more complex and requires future work.
>
> We also have fixed the typos and uploaded a rebuttal revision (specifically Line 236, line 303, line 305 and line 309 in the revision). Appendix is in the supplementary materials, as is required by Neurips submission guideline.

---

> > ### Comment · Area_Chair_rn2L · 2022-08-09
> > **Reviewer cok9, please respond to the rebuttal of paper 11375**
> >
> > Reviewer cok9,
> >
> > You have yet to review the response from the authors of paper 11375.
> > As the deadline is today, please read the response asap and indicate whether your concerns were addressed.
> >
> > All the best,
> >
> > The AC

---

> ### Author Response · Authors · 2022-08-09
> **Response to Reviewer cok9**
>
> To summarize, we emphasize that UnfoldML uses pre-trained models to create a dynamic cost aware 2D pipelines and model architecture searching in NAS is beyond the scope of this paper. We also highlight that searching the entire cost-AUC tradeoff space is not needed for configuring the UnfoldML prediction pipeline. Given the upcoming OpenReview deadline, we’d like to confirm with the reviewer whether our response addressed their concerns and if there are any additional questions we could clarify.

---

### Official Review · Reviewer_f34b · 2022-07-11

**Rating:** 4
**Confidence:** 5
**Soundness:** 2 fair
**Presentation:** 1 poor
**Contribution:** 2 fair

**Summary:**

Sequential classification problems, where there is an existing hierarchy among the classes with one class transforming into the next set of classes, are commonly observed in many clinical settings of disease progression. Generally, the state-of-the-art methods in these settings treat all the stages as a multi-class problem without considering the hierarchy. This paper proposes a model unfolding strategy that trains a series of models of increasing complexity for each stage and progressively moves a data sample to next stages only when the first stage is detected. The proposed model leads to a reduction in both memory as well as inference time costs instead of running a single complex classifier. A Sepsis-Septic Shock prediction task and subcategory detection in CIFAR-100 dataset is used to evaluate the proposed method and baselines.

**Questions:**

Please refer to above section.

**Strengths And Weaknesses:**

Overall this paper studies an important area of model decomposition. The key concern in the paper is experimental results. Although the model unfolding idea is interesting, the baselines used are not explained adequately. Further analysis based on the algorithm's robustness and ease of use need to be done.
- An important component of the proposed methodology is the measured confidence criterion of a model. Accurate estimation of the uncertainty of a deep learning model has been an extensive area of research. This paper lacks the comparison with other uncertainty-quantification methods like MC-dropout, Bayesian approximations, etc., which are more state-of-the-art. Additionally, the algorithms depend on the reliability of the individual stage classifiers, and thus under-trained parts classifiers can lead to faulty signals for gating operations.
- In addition, the baselines are not adequately explained, which makes it very hard to judge the comparisons properly. For example, the terms LSTM-C, LSTM-A, Multi-class LSTM (Rec.) has been used without proper explanations.
- The writing in the paper lacks clarity which makes it difficult to understand. A few examples are listed below:
1. Line 156, the process of calculating the model's confidence is absent.
2. The optimization of Objective II (Sec. 3.2.3) is not explained properly. The paper is not self-contained, with most of the discussion deferred to the Appendix.
3. Experimental settings are also not explained in the paper (Sec. 4.1). What are the features?
4. The Baseline section (line 280) is poorly written, with the baselines not matching the terms being used in the tables and figures.
5. Table 4.1.2 (line 314) is absent in the paper.
6. Several other typos need to be fixed.

---

> ### Author Response · Authors · 2022-08-02
> **Thanks for your feedback.**
>
> Thanks for your feedback. Please see our response per each of your concerns below:
>
> 1. *“An important component of the proposed methodology is the measured confidence criterion of a model. Accurate estimation of the uncertainty of a deep learning model has been an extensive area of research. This paper lacks the comparison with other uncertainty-quantification methods like MC-dropout, Bayesian approximations, etc., which are more state-of-the-art”*
>
> First, there is a misunderstanding. The confidence criterion is not a key component in the proposed method. The focus of this paper is gate parameter learning for DAG-decompositions of monolithic multi-class classifiers.
>
> Second, the suggested uncertainty-quantification methods like MC-dropout/Bayesian approximation do not suit our cost-aware prediction pipeline. These methods require multiple runs at inference time in order to get uncertainty estimation, so they significantly increase the inference time (i.e., temporal cost).
>
> We made a choice of confidence criteria to be post-processing scores that can be derived from predicted probabilities in close forms, to avoid using any additional inference time. This complies with our goal of minimizing the spatio-temporal cost of inference. We experimented with ‘max of probability’, ‘entropy’, ‘entropy of expected’ and ‘mutual information’ (see Appendix 5.5), and select ‘entropy of expected’ to report in the main paper because it outperforms the other three as shown in Appendix Figure 3.
>
> 2. *“the algorithms depend on the reliability of the individual stage classifiers, and thus under-trained parts classifiers can lead to faulty signals for gating operations.”*
>
> We do not see why the single stage classifiers can be under-trained if the multi-stage classifiers were not. The amount of positive cases labeled for each class/stage do not change regardless if we unfold the multi-task classification or not. The imbalance problem between the positive and negative cases are addressed in the training process of single stage binary classifiers by oversampling the positive cases in each batch. Table 2 and 4 show that UnfoldML (Rec.) achieves similar end-to-end AUC as the most accurate multi-task classifier but with significantly lower cost, which clearly demonstrates that unfold classifiers are not under-trained.
>
> 3. *“ the baselines are not adequately explained, which makes it very hard to judge the comparisons properly. For example, the terms LSTM-C, LSTM-A, Multi-class LSTM (Rec.) has been used without proper explanations.”*
>
> We have provided detailed explanations of ‘-C’, ‘-A’ and ‘(Rec.)’ in the caption of Table 2. We also have further explained ‘(Rec)’ in line 317-319.
>
> 4. *“Line 156, the process of calculating the model's confidence is absent.”*
>
> For the sake of space, we moved the confidence calculation to Appendix 5.5. We will summarize them and put them back in the final version for clarification.
>
> 5. *“The optimization of Objective II (Sec. 3.2.3) is not explained properly. The paper is not self-contained, with most of the discussion deferred to the Appendix.”*
>
> The optimization of Sub-Objective 2 has been properly explained in Line 241-246. Discussions that do not affect the main results are moved to Appendix due to page limit.
>
> 6. *“Experimental settings are also not explained in the paper (Sec. 4.1). What are the features?”*
>
> Feature sets used for model training are between line 271-274 in Sec. 4.1. Training details of the models in the zoo are written in line 265-271 in Sec. 4.1. Details on how to extract each feature are described in data preparation in Appendix 5.2.
>
> 7. *“The Baseline section (line 280) is poorly written, with the baselines not matching the terms being used in the tables and figures.”*
>
> We do not identify any unmatched terms regarding the baselines among the tables/figures and the Baseline section. Could you please clarify?
>
> 8. *“Table 4.1.2 (line 314) is absent in the paper.”*
>
> Cross reference is now fixed in the rebuttal revision (line 309 in the revision). It should be Table 1. We also fix some other typos in the revision.

---

> > ### Comment · Reviewer_f34b · 2022-08-09
> > **Thank you for the response**
> >
> > Thanks to the authors to take time to carefully answer the questions. I still feel there is a need to carefully analyze what policy is being chosen by the gating functions. There is no explainability of the model predictions given the primary experiments are being done on a healthcare dataset. I would move the score to 4: Borderline reject. However, I feel this is an interesting work and has the potential to have impact in reducing spatio-temporal cost of training and inference.

---

> > > ### Author Response · Authors · 2022-08-09
> > > **clarifying the main focus of the paper**
> > >
> > > We have carefully defined the evaluation metrics, i.e., the cost-AUC tradeoff quantified by the area under convex hull in Table 1 and early-hour prediction in Table 2, in the paper and have shown that UnfoldML outperforms the baselines in these evaluations quantitatively. The suggested qualitative analysis, e.g., demonstration on policy selection or explanation on model predictions, are not the main focus of our experiments. So we include them as Section 5.6 in the Appendix.
> > >
> > > We do appreciate the reviewer's interest in explainability, but this goal is just not in scope. We're glad to see the reviewer's appreciation for the **main focus** of the paper , _namely reducing spatio-temporal cost of training and inference_. Our key success metrics are explicitly defined w.r.t. this explicitly stated goal and thoroughly evaluated w.r.t those success metrics, outperforming baselines by more than an order of magnitude.

---

> > > > ### Author Response · Authors · 2022-08-09
> > > > **score not yet updated**
> > > >
> > > > p.s. if you're planning to increase the score, could you please do that? We don't see it reflected on the scoreboard on our side. Thanks.

---

> ### Author Response · Authors · 2022-08-09
> **Response to Reviewer f34b**
>
> To summarise,  we emphasize that uncertainty quantification is NOT the contribution of our work. We further justify why other uncertainty quantification methods (pointed out by the reviewer)  are not relevant for UnfoldML. Lastly, we highlight that unfold classifiers are not under-trained with proper reasoning.  Given the upcoming OpenReview deadline, we’d like to confirm with the reviewer whether our response addressed their concerns and if there are any additional questions we could clarify.

---

> > ### Author Response · Authors · 2022-08-09
> > **reviewer confidence**
> >
> > The reviewer states "absolute certainty" about their assessment, but there are several important points of confusion that negatively affected the score. We do appreciate the reviewer's careful and in-depth reading of the paper and we invite a dialog to clarify any points of confusion that may remain after our rebuttal.

---

### Author Response · Authors · 2022-08-02
**General Comment**

We thank the reviewers for their insightful comments. We reiterate our  key contributions. UnFoldML -
1. A novel 2D prediction pipeline that decomposes (unfolds) multi-stage prediction classifiers by leveraging happens-before relationships between stages.
2. Formulates two gating algorithms (hard and soft) to enable traversing latency/cost trade-off space and to find optimal points in that space.
3. Reduces inference cost (spatio-temporal) by an order of magnitude.

We highlight key clarifications from the reviews -
 1. **UnFoldML doesn't propose a new uncertainty estimation**
The goal of UnFoldML is not to propose a novel method for uncertainty estimation.The focus of our method is on how to serve the models that have already been trained  rather than train the models in the zoo. Furthermore,  UnFoldML reduces inference cost of the prediction workflows and achieves a better in the latency/cost trade-off space by decomposing prediction workflows onto multiple stages.

2. **Theoretical Explanation**
We agree with reviewers that adding a more detailed theoretical explanation (in addition to the mathematical formulation in Sec3.1 and Sec3.2) would further strengthen the paper. The optimization objective targeted by UnFoldML is not convex. Provable optimality is hard to achieve with any heuristic. Our two proposed heuristics are designed to find cost-effective solutions in the latency/cost trade-off space in UnFoldML as a demonstration of feasibility. Results may be further improved with better heuristics.

3. **Baseline** We compare UnFoldML with a baseline which is "not-decomposed" i.e multi-class classifier. We demonstrate that UnFoldML achieves an order-of-magnitude cost reduction compared to multi-class classifiers.  IDKCascade is another, more recent baseline that   traverses latency/accuracy trade-off space for a single stage of prediction.

---

### Meta-Review · Area_Chair_rn2L · 2022-08-31

**Recommendation:** Accept
**Confidence:** Less certain

**Metareview:**

The paper presents a method for transforming a multi-class multimodal classifier into a multi-stage classifier, increasing the efficiency at runtime and only using the necessary modalities for prediction. Most reviewers agree that this is an important problem for the community and has a high potential for impact. Reviewer f34b raised a question about uncertainty quantification, which the authors clarified successfully. The authors also introduced an appendix for policy selection and explanation on model predictions at the reviewer's request -- as these are tangential to the paper, the effort is appreciated. The authors also answered the questions about the optimization problem asked by reviewer cok9 - the reviewer did not comment on the author response, however the answers seem pertinent.
An remaining issue is the lack of theoretical analysis of the work, though the experiments do support the authors' claims about the cost reduction. Reviewers 5CL8 and qEBg issued positive comments w.r.t the paper's strengths, specifically the real world experiments, the reduction in cost and the exploration of the cost-AUC trade-off space.

**Award:**

No

---

### Decision · Program_Chairs · 2022-09-14

Accept